# Patterns of multimorbidity and demographic profile of latent classes in a Danish population—A register-based study

**Sanne Pagh Møller***, **Bjarne Laursen, Caroline Klint Johannesen, Janne S. Tolstrup, Stine Schramm**

National Institute of Public Health, University of Southern Denmark, Copenhagen, Denmark

* sapm@sdu.dk

## Abstract

### Background

Multimorbidity is an increasing public health concern and is associated with a range of further adverse outcomes. Identification of disease patterns as well as characteristics of populations affected by multimorbidity is important for prevention strategies to identify those at risk.

### Aim

The aim of the study was to identify and describe demographic characteristics of multimorbidity classes in three age groups (16–44 years, 45–64 years, and 65+ years).

### Methods

Based on register information on 47 chronic diseases and conditions, we used latent class analysis to identify multimorbidity classes in a random sample of the Danish population (n = 470,794). Information on sociodemographic characteristics (age, sex, region of origin, educational level, employment status, and marital status) was obtained from registers and linked to the study population. Age- and sex-adjusted multinomial logistic regression models were used to examine associations between multimorbidity classes and sociodemographic characteristics.

### Results

We identified seven classes among individuals in the age groups 45–64 years and 65+ years and five classes in the age group 16–44 years. Overall, the classes were similar in the three age groups, but varied in size, i.e. the class 'No or few diseases' was larger in the younger age group. The class 'Many diseases' (a class with both somatic diseases and mental illnesses) was only seen in individuals aged 45–64 years and 65+ years. There were social inequalities in odds of belonging to the multimorbidity classes compared to the healthier class. These social inequalities varied but were especially strong in the classes named 'Many diseases' and 'Mental illness, epilepsy'.

**Data Availability Statement:** Due to Danish law, the confidential health care data used in this study can only be accessed through Statistics Denmark. Access is granted upon request to applicants who

fulfill the necessary criteria. Data access requests can be sent directly to Statistics Denmark via the following email address: databanker@dst.dk.

**Funding:** The authors received no specific funding for this work.

**Competing interests:** The authors have declared that no competing interests exist.

## Conclusion

The results of the study suggest that there are social inequalities in multimorbidity but that these inequalities are not universal to all types of multimorbidity. This supports that multimorbidity is diverse and should be prevented and treated accordingly.

## Background

Worldwide increases in life expectancy together with improvements in disease management, has led to increasing prevalence of chronic diseases, as well as co-occurring chronic disease within the same individuals also known as multimorbidity [1]. Multimorbidity, often defined as having more than one chronic disease, is associated with premature death, hospitalization, polypharmacy, reduced quality of life, and a substantial economic burden for health systems [2]. Thus, multimorbidity is a public health burden today and is estimated to increase in the future. Knowledge about multimorbidity and the people who live with multimorbidity is therefore an important contribution to disease management and prevention strategies.

Studies on multimorbidity vary substantially in study population and the applied methods to estimate multimorbidity, as well as the number of diseases, types of diseases, and definition of diseases included [3, 4]. For this reason, reported estimates of prevalence of multimorbidity vary greatly in the literature. One method previously used is latent class analysis (LCA) which is used to study phenomena that cannot be directly measured. With LCA it is possible to divide a population into categories of an unobserved variable [5]. If the LCA is based on information on diseases this will result in categories that represent groups of individuals with different disease profiles. This technique has been used in previous studies identifying patterns of multimorbidity in different populations [6–9].

As the prevalence of multimorbidity increases with age [10, 11], many studies on multimorbidity have focused on older populations. However, multimorbidity is not restricted to the elderly, and as patterns of multimorbidity are likely to change over the life course, it is important to study multimorbidity across age groups [12]. Previous studies have shown that mental health conditions were more common in multimorbidity in the younger age groups compared to the older [13], and that age distributions within different groups of multimorbidity vary substantially [6]. In order to inform prevention strategies for multimorbidity, it is also relevant to gain knowledge on multimorbidity not only among the elderly but also in younger populations.

Generally, sociodemographic characteristics of individuals with a certain pattern of multimorbidity can contribute with important knowledge about the different patterns of multimorbidity which may influence the prevention strategies for multimorbidity and consequences hereof. Studies have shown that sex, age, and socioeconomic status are all associated with the pattern of multimorbidity [3, 11, 14]. This has also been shown in a study of the general Danish population aged 16 years and older [6]. Some studies have also shown that pattern of multimorbidity is associated with outcomes such as mortality and health-related quality of life [6, 15, 16]. However, studies of multimorbidity have primarily been based on self-reported information on selected diseases, and therefore it is relevant to study the pattern of multimorbidity using register-based information. This source ensures that the included diseases have been clinically validated. The aim of this study was to describe the pattern of multimorbidity and the demographic characteristics of multimorbidity in an adult population in Denmark stratified into three age groups using register-based data on 47 diseases.

## Methods

The study population consisted of a 10% random sample of all persons aged 16 years or older with permanent residence in Denmark on January 1st 2017 (n = 470,794). This was drawn from register the Danish Civil Registration System which includes all Danish residents [17], and the deviation in age/sex distribution of the sample from the total population using 10 years age group was less than 1% up to 79 years. For all analyses, the study population was divided into three age groups (16–44 years, 45–64 years, and 65+ years). These age groups were defined as 1) a young age group based on individuals who would be expected to have no or few diseases (16–44 years); 2) a middle-aged group based on individuals who would to some degree be expected to be disease free (45–64 years); and 3), an elderly group based on individuals who have mainly retired from the work force and would be expected to be affected by one or more diseases (65+ years). Linkage of the study population with national registers was possible using a unique personal identification number available to all Danish residents. Information on sex, age, place of birth, and marital status of the study population was obtained from the Civil Registration System, which contains information on all residents in Denmark [17]. Information on highest educational level attained was obtained by linking the cohort to the Population's Education Register [18], and information on employment status was obtained from the Register-based Labour Force Statistics [19].

Pattern of multimorbidity was based on information on 47 chronic diseases (see S1 Table). Identification of diseases was made by applying diagnoses and purchase of prescribed medications primarily based on algorithms applied in previous studies [20, 21] (See S1 Table for algorithms). Information on diagnoses and prescriptions was obtained through linkage of the study population with the Danish National Patient Register (NPR), The Danish National Psychiatry Central Research Register (PCRR), and the Danish National Prescription Registry (DNPR). NPR contains information on all contacts to the secondary health care system in Denmark [22], PCRR contains information on all contacts to mental hospitals and psychiatric departments [23], and DNPR contains information on redeemed prescription drugs sold in Danish pharmacies [24].

Individuals with one or more of the 47 diseases on January 1st 2017 were identified. As it cannot be expected that all chronic diseases have a lifelong presence, it was decided that only diseases that led to hospital contact within the previous 10 years were included. For some diseases the contact was required to be within the previous five years, as lack of contact within five years would indicate that the particular disease had diminished (see S1 Table). The decision on time frame for specific diseases was based on previous work by Hvidberg et al 2016 [21]. Diseases identified through medicine prescription were only included if a minimum of two prescription had been redeemed within the previous two years.

### Statistical analysis

Applied methods to assess multimorbidity include counting the number of diseases, estimations of the observed versus the expected prevalence of the disease combinations, and techniques such as K-means, factor analysis, or LCA to estimate the patterns of multimorbidity [3, 4, 25]. In this study, LCA was used to assess multimorbidity as this approach is exploratory and does not entail a priori assumptions about patterns of multimorbidity. Also, in LCA the latent variable is considered to be a discrete variable as opposed to a continuous variable. However, we do not consider the LCA approach to be superior to other methods.

The pattern of multimorbidity was assessed using LCA with the 47 diseases as observed indicators. For each age group and number of classes the estimation was run up to 100 times to ensure that the optimal solution based on likelihood was found. The optimal number of

latent classes was based on the following criteria: model information based criteria (namely, BIC and AIC); acceptable fit of the model to the data; presence of distinct classes; identified classes can be meaningfully interpreted; and inclusion of at least 1% of the population in the smallest class. Nine was the maximum number of clusters evaluated. The AIC and BIC values continued to decrease as more latent classes were added to the model (S5 Table), but models with distinguishable classes were prioritised. Individuals were subsequently assigned to the latent class that they had the highest probability of membership to. Naming of classes was based on diseases identified as particularly prevalent in one class compared to the other classes. LCA was carried out separately for the three age groups 16–44 years, 45–64 years, and 65 + years to avoid identification of primarily age dependent classes.

Demographic characteristics of the identified classes were compared using chi-squared tests for significance for categorical variables and Kruskal-Wallis tests for significance for age. The demographic characteristics included age, sex, country of origin (Danish; Other Western; Non-Western), educational level (Missing; Elementary school, Short education; Medium/long education), employment status (Working; Unemployed; Sick leave etc.; Early retirement pension; Retired; Student; Other (generally low attachment to the labour market but not receiving social payments)), and marital status (Unmarried; Married). To asses associations between the demographic variables and the different classes, we applied a multinomial logistic regression model, which was adjusted for age and sex.

Bias may be introduced from assigning individuals to the class with the highest probability, and therefore a sensitivity analysis was conducted in which the multinomial logistic regression included weights based on the probability of belonging to this class. This analysis was based on data from the age group 45–64 years.

All analyses were performed in SAS version 9.4 (SAS Institute Inc, Cary, North Carolina, USA). The SAS procedure PROC LCA was used for the latent class estimation [26].

Ethical permission for scientific studies carried out in Denmark is necessary only when they include biological samples, such as blood or tissue, or information from medical records. Therefore, ethical permission was not necessary for this study. All data were fully anonymised before access.

## Results

Table 1 shows characteristics of the study population stratified in the three age groups. There were more women than men in the age group 65+ years (54%), whereas about 49% were women in the two other age groups. In the age group 16–44 years, 11.9% were not born in Denmark compared to the older age groups (2.0% and 6.5% for 65+ and 45–64 years of age, respectively). Elementary school as the highest attained educational level was more common in the oldest age group compared to the younger, and employment status and marital status also varied between the age groups.

The LCA resulted in the identification of seven classes in the 65+ years age group (Table 2). The seven classes were named: 'No or few diseases' (46.3% of the age group), 'Diabetes cholesterol' (25.2% of the age group) 'Heart disease' (8.3% of the age group) 'Back disease, asthma, allergy' (6.3% of the age group) 'Many diseases' (5.4% of the age group) 'COPD, cancer, liver disease' (4.4% of the age group), and 'Mental illness, epilepsy' (4.2% of the age group). Seven classes were also identified in the age group 45–64 years (Table 2). These were named: 'No or few diseases' (69.4%), 'Diabetes, cholesterol' (11.4%), 'Bone and joint diseases' (9.9%), 'Mental illness, epilepsy' (2.9%), 'Heart diseases' (2.3%), 'Many diseases' (2.1%), and 'Asthma, allergy' (2.1%). In the age group 16–44 years, five classes were identified (Table 2) and named 'No or few diseases' (90.3%), 'Bone and joint diseases' (3.8%), 'Mental illness, epilepsy' (3.5%),

**Table 1. Characteristics of the three study populations.**

| | 65+ years (n = 109.318) | 45–64 years (n = 151.870) | 16–44 years (n = 209.606) |
|---|---|---|---|
| | n (%) | n (%) | n (%) |
| **Age,** mean (sd) | 74.4 (7.2) | 54.1 (5.7) | 30.0 (8.5) |
| **Sex** | | | |
| Men | 50,333 (46.0) | 76,047 (50.1) | 106,537 (50.8) |
| Women | 58,985 (54.0) | 75,823 (49.9) | 103,069 (49.2) |
| **Country of origin** | | | |
| Danish | 101,114 (95.2) | 136,726 (90.0) | 169,982 (81.1) |
| Other Western | 3,075 (2.8) | 5,299 (3.5) | 14,631 (7.0) |
| Non-Western | 2,129 (2.0) | 9,844 (6.5) | 24,985 (11.9) |
| **Educational level** | | | |
| Missing | 2,704 (2.5) | 4,941 (3.3) | 19,900 (9.5) |
| Elementary school | 39,883 (36.5) | 32,315 (21.3) | 61,66 (29.4) |
| Short education[1] | 46,042 (42.1) | 74,647 (49.2) | 81,961 (39.1) |
| Medium/long education[2] | 20,689 (18.9) | 39,967 (26.3) | 46,145 (22.0) |
| **Employment status** | | | |
| Working | 8,880 (8.1) | 114,032 (75.1) | 121,758 (58.1) |
| Unemployed | 181 (0.2) | 10,301 (6.8) | 14,851 (7.1) |
| Sick leave etc.[3] | - | 1,362 (0.9) | 3,103 (1.5) |
| Early retirement pension | 643 (0.6) | 15,671 (10.3) | 4,429 (2.1) |
| Retired | 99,546 (91.1) | 5,866 (3.9) | - |
| Other | 68 (0.1) | 4,638 (3.1) | 20,089 (9.6) |
| Student | - | - | 45,376 (21.7) |
| **Marital status[4]** | | | |
| Unmarried | 47,028 (43.0) | 59,524 (39.2) | 150,229 (71.7) |
| Married | 62,290 (57.0) | 92,346 (60.8) | 59,377 (28.3) |

[1] Completed high school, vocational school, or short tertiary education

[2] Completed medium or long tertiary education (>3 years)

[3] Includes individuals on sick leave, maternity leave, or other types of leave related to for example training.

[4] Marital status in the year 2017

'Asthma, allergy' (1.4%), and 'Diabetes, heart diseases' (1.0%). Fig 1 shows the prevalence the most prevalent diseases and of diseases with large variations in prevalence between the classes. The prevalence of all diseases in all classes are presented in S2–S4 Tables.

The demographic profiles of the classes in the age group 65+ years are presented in Table 3. Results from the multinomial logistic regression model, with the class 'No or few diseases' as reference, showed that compared to men, female sex was associated with higher odds of being in the classes: 'Back disease, asthma, allergy' (OR = 1.9; 95%CI:1.8–2.0), 'COPD, cancer, liver disease' (OR = 1.6; 95%CI:1.5–1.8), and 'Mental illness, epilepsy' (OR = 1.3; 95%CI:1.2–1.4), whereas being female was associated with lower odds of being in the classes: 'Heart diseases' (OR = 0.6; 95%CI:0.5–0.6), 'Diabetes, cholesterol' (OR = 0.7; 95%CI:0.7–0.8), and 'Many diseases' (OR = 0.8; 95%CI:0.8–0.9). Non-Western origin was associated with higher odds of being in the classes 'Diabetes, cholesterol' (OR = 1.7; 95%CI:1.5–1.9), 'Back disease, asthma, allergy' (OR = 1.3; 95%CI:1.1–1.5), and 'Many diseases' (OR = 1.9; 95%CI:1.6–2.2). The demographic profile of the seven classes identified in the age group 45–64 years can be seen in Table 4. This shows that with the class 'No or few diseases' as reference, female sex was associated with higher odds of being in the classes 'Bone, joint diseases', 'Mental illness, epilepsy',

**Table 2. Class sizes and no. of diseases in classes among the age groups 65+, 45–64, and 16–44 years.**

| 65+ years | | | | | | | |
|---|---|---|---|---|---|---|---|
| Mean no. of diseases: 2.8 | 'No or few diseases' 46.3% (n = 50,605) | 'Diabetes, cholesterol' 25.2% (n = 27,502) | 'Heart diseases' 8.3% (n = 9,065) | 'Back disease, asthma allergy' 6.3% (n = 6,866) | 'Many diseases' 5.4% (n = 5,884) | 'COPD, cancer, liver disease' 4.4% (n = 4,756) | 'Mental illness, epilepsy' 4.2% (n = 4,640) |
| | % | % | % | % | % | % | % |
| No. of diseases | | | | | | | |
| 0–1 | 68.5 | 0.0 | 0.0 | 0.0 | 0.0 | 0.0 | 0.0 |
| 2 | 25.0 | 22.0 | 9.0 | 1.5 | 0.0 | 6.8 | 5.6 |
| 3 | 6.1 | 32.9 | 18.7 | 29.0 | 0.0 | 34.3 | 19.7 |
| 4 | 0.4 | 26.1 | 23.4 | 31.8 | 0.1 | 28.2 | 21.3 |
| >4 | 0.0 | 19.1 | 48.9 | 37.8 | 99.9 | 30.7 | 53.4 |
| Mean no. of diseases (sd) | 1.1 (0.9) | 3.5 (1.1) | 4.5 (1.5) | 4.3 (1.3) | 7.7 (1.6) | 4.0 (1.3) | 4.8 (1.8) |
| **45–64 years** | | | | | | | |
| Mean no. of diseases: 1.4 | 'No or few diseases' 69.4% (n = 105,416) | 'Diabetes, cholesterol' 11.4% (n = 17,349) | 'Bone-, joint diseases' 9.9% (n = 15,002) | 'Mental illness, epilepsy' 2.9% (n = 4,337) | 'Heart diseases' 2.3% (n = 3,439) | 'Many diseases' 2.1% (n = 3,201) | 'Asthma, allergy' 2.1% (n = 3,126) |
| | % | % | % | % | % | % | % |
| No. of diseases | | | | | | | |
| 0–1 | 91.7 | 10.5 | 2.0 | 0.0 | 0.0 | 0.0 | 0.0 |
| 2 | 8.3 | 29.4 | 41.7 | 25.2 | 17.7 | 0.0 | 31.3 |
| 3 | 0.1 | 31.2 | 35.6 | 29.1 | 24.0 | 0.1 | 40.3 |
| 4 | 0.0 | 19.1 | 13.7 | 22.4 | 23.7 | 5.3 | 16.6 |
| >4 | 0.0 | 9.8 | 7.2 | 23.3 | 34.6 | 94.6 | 11.8 |
| Mean no. of diseases (sd) | 0.5 (0.6) | 2.9 (1.2) | 2.8 (1.0) | 3.6 (1.4) | 4.0 (1.5) | 6.7 (1.7) | 3.1 (1.1) |
| **16–44 years** | | | | | | | |
| Mean no. of diseases: 0.5 | 'No or few diseases' 90.3% (n = 189,202) | 'Bone-, joint diseases' 3.8% (n = 7,962) | 'Mental illness, epilepsy' 3.5% (n = 7,378) | 'Asthma, allergy' 1.4% (n = 2,985) | 'Diabetes, heart diseases' 1.0% (n = 2,079) | | |
| | % | % | % | % | % | | |
| No. of diseases | | | | | | | |
| 0–1 | 97.2 | 4.7 | 0.5 | 0.0 | 0.0 | | |
| 2 | 2.8 | 56.0 | 44.9 | 68.0 | 32.8 | | |
| 3 | 0.0 | 27.2 | 29.9 | 23.3 | 31.0 | | |
| 4 | 0.0 | 8.1 | 13.7 | 6.4 | 17.5 | | |
| >4 | 0.0 | 4.0 | 11.1 | 2.3 | 18.8 | | |
| Mean no. of diseases (sd) | 0.3 (0.5) | 2.5 (0.9) | 3.0 (1.2) | 2.4 (0.7) | 3.4 (1.5) | | |

'Many diseases', and 'Asthma, allergy', whereas females had lower odds of being in the classes 'Diabetes, cholesterol' and 'Heart diseases'. Other Western origin was associated with lower odds of belonging to most multimorbidity classes, whereas Non-Western origin was associated with higher odds of belonging to most multimorbidity classes. Table 5 describes the sociodemographic characteristics of the five identified classes in the age group 16–44 years. It shows that with the class 'No or few diseases' as reference, odds of belonging to any of the multimorbidity classes except 'Asthma, allergy' increased with age. Female sex was associated with higher odds of belonging to all classes except 'Diabetes, heart diseases', which women had

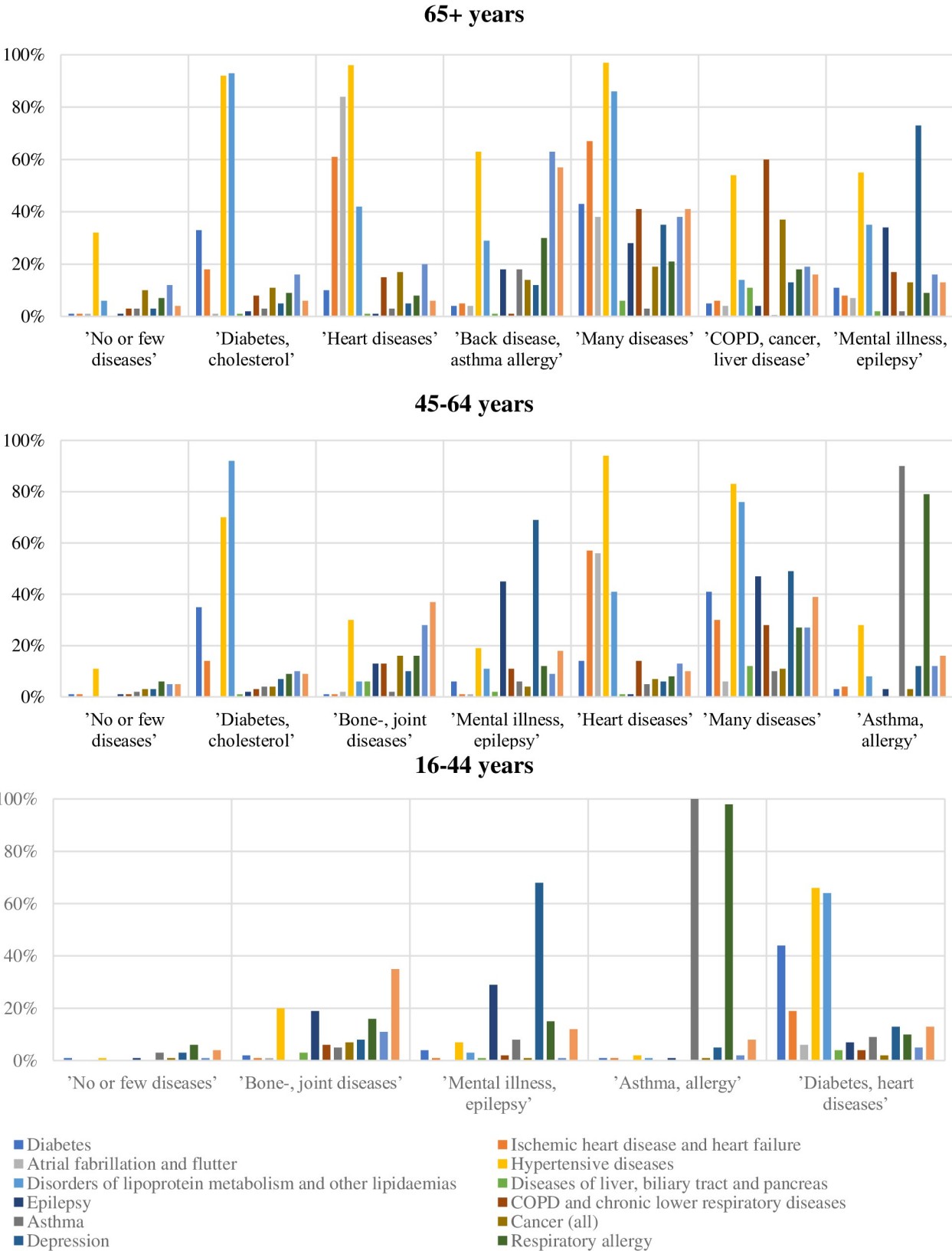

**Fig 1. Prevalence of selected diseases in classes among the age groups 65+, 45–64, and 16–44 years.**

**Table 3. Demographic profile of individuals by assigned classes in the age group 65+ years.**

| | 'No or few diseases' 46.3% § (n = 50,605) | | 'Diabetes, cholesterol' 25.2% (n = 27,502) | | 'Heart diseases' 8.3% (n = 9,065) | | 'Back disease, asthma, allergy' 6.3% (n = 6,866) | | 'Many diseases' 5.4% (n = 5,884) | | 'COPD, cancer, liver disease' 4.4% (n = 4,756) | | 'Mental illness, epilepsy' 4.2% (n = 4,640) | | p |
|---|---|---|---|---|---|---|---|---|---|---|---|---|---|---|---|
| | % | OR | % | OR [95%CI][1] | % | OR [95%CI][1] | % | OR [95%CI][1] | % | OR [95%CI][1] | % | OR [95%CI][1] | % | OR [95%CI][1] | |
| **Age** | | | | | | | | | | | | | | | *** |
| 65–74 § | 67.5 | 1.0 | 57.9 | 1.0 | 37.2 | 1.0 | 55.8 | 1.0 | 42.3 | 1.0 | 53.5 | 1.0 | 44.2 | 1.0 | |
| 75–84 | 24.8 | 1.0 | 33.1 | 1.6 [1.5;1.6] | 39.1 | 2.9 [2.8;3.1] | 32.8 | 1.6 [1.5;1.7] | 41.0 | 2.7 [2.5;2.8] | 34.4 | 1.7 [1.6;1.8] | 33.4 | 2.0 [1.9;2.2] | |
| 85+ | 7.7 | 1.0 | 9.1 | 1.5 [1.4;1.5] | 23.7 | 6.1 [5.8;6.5] | 11.4 | 1.6 [1.5;1.8] | 16.8 | 3.6 [3.3;3.9] | 12.1 | 1.9 [1.7;2.0] | 22.5 | 4.3 [4.0;4.7] | |
| **Sex** | | | | | | | | | | | | | | | *** |
| Men § | 45.1 | 1.0 | 52.3 | 1.0 | 56.0 | 1.0 | 29.5 | 1.0 | 48.0 | 1.0 | 32.4 | 1.0 | 36.0 | 1.0 | |
| Women | 54.9 | 1.0 | 47.7 | 0.7 [0.7;0.8] | 44.1 | 0.6 [0.5;0.6] | 70.5 | 1.9 [1.8;2.0] | 52.0 | 0.8 [0.8;0.9] | 67.6 | 1.6 [1.5;1.8] | 64.0 | 1.3 [1.2;1.4] | |
| **Country of origin** | | | | | | | | | | | | | | | *** |
| Danish § | 95.2 | 1.0 | 94.9 | 1.0 | 96.1 | 1.0 | 95.2 | 1.0 | 94.7 | 1.0 | 96.2 | 1.0 | 95.5 | 1.0 | |
| Other Western | 3.1 | 1.0 | 2.4 | 0.8 [0.7;0.9] | 2.7 | 0.9 [0.8;1.0] | 2.8 | 0.9 [0.8;1.0] | 2.7 | 0.9 [0.7;1.0] | 2.6 | 0.8 [0.7;1.0] | 3.2 | 1.0 [0.9;1.2] | |
| Non-Western | 1.7 | 1.0 | 2.7 | 1.7 [1.5;1.9] | 1.3 | 0.9 [0.7;1.1] | 2.0 | 1.3 [1.1;1.5] | 2.7 | 1.9 [1.6;2.2] | 1.2 | 0.8 [0.6;1.0] | 1.3 | 0.9 [0.7;1.2] | |
| **Marital status** | | | | | | | | | | | | | | | *** |
| Unmarried | 39.7 | 1.0 | 41.1 | 1.0 [1.0;1.1] | 47.0 | 1.1 [1.0;1.1] | 44.3 | 0.9 [0.9;1.0] | 52.2 | 1.4 [1.3;1.4] | 51.6 | 1.3 [1.2;1.4] | 61.0 | 1.8 [1.7;1.9] | |
| Married § | 60.3 | 1.0 | 58.9 | 1.0 | 53.0 | 1.0 | 55.8 | 1.0 | 47.8 | 1.0 | 48.4 | 1.0 | 39.0 | 1.0 | |

***: p<0.001

§: reference group; OR: Odds ratio compared to the reference group of being in a multimorbidity class compared to the reference class; *p*: Chi²-test for univariate association between demographic variable and classes

[1]Adjusted for age and sex.

lower odds of belonging to compared to men. Non-Danish origin was associated with lower odds of belonging to almost all multimorbidity classes compared to Danish origin.

Sensitivity analyses weighted for the probability of belonging to a class in the age group 45–64 years showed similar results (S9 Table).

**Table 4. Demographic profile of individuals by assigned classes in the age group 45–64 years.**

| | 'No or few diseases' 69.4% § (n = 105,416) | | 'Diabetes, cholesterol' 11.4% (n = 17,349) | | 'Bone-, joint diseases' 9.9% (n = 15,002) | | 'Mental illness, epilepsy' 2.9% (n = 4,337) | | 'Heart diseases' 2.3% (n = 3,439) | | 'Many diseases' 2.1% (n = 3,201) | | 'Asthma, allergy' 2.1% (n = 3,126) | | p |
|---|---|---|---|---|---|---|---|---|---|---|---|---|---|---|---|
| | % | OR | % | OR [95%CI][1] | % | OR [95%CI][1] | % | OR [95%CI][1] | % | OR [95%CI][1] | % | OR [95%CI][1] | % | OR [95%CI][1] | |
| **Age** | | | | | | | | | | | | | | | *** |
| 45–54 § | 59.7 | 1.0 | 32.5 | 1.0 | 43.6 | 1.0 | 58.7 | 1.0 | 28.9 | 1.0 | 31.6 | 1.0 | 57.2 | 1.0 | |
| 55–64 | 40.3 | 1.0 | 67.5 | 3.1 [3.0;3.2] | 56.4 | 1.9 [1.8;2.0] | 41.3 | 1.0 [1.0;1.1] | 71.1 | 3.7 [3.4;4.0] | 68.4 | 3.2 [3.0;3.4] | 42.8 | 1.1 [1.0;1.2] | |
| **Sex** | | | | | | | | | | | | | | | *** |
| Men § | 51.2 | 1.0 | 59.5 | 1.0 | 35.0 | 1.0 | 42.2 | 1.0 | 65.3 | 1.0 | 43.5 | 1.0 | 34.4 | 1.0 | |
| Women | 48.8 | 1.0 | 40.5 | 0.7 [0.7;0.7] | 65.0 | 1.9 [1.9;2.0] | 57.8 | 1.4 [1.3;1.5] | 34.7 | 0.5 [0.5;0.6] | 56.5 | 1.3 [1.3;1.4] | 65.6 | 2.0 [1.9;2.2] | |
| **Country of origin** | | | | | | | | | | | | | | | *** |
| Danish § | 90.3 | 1.0 | 88.8 | 1.0 | 90.9 | 1.0 | 84.1 | 1.0 | 92.5 | 1.0 | 87.7 | 1.0 | 89.7 | 1.0 | |
| Other Western | 3.9 | 1.0 | 2.5 | 0.7 [0.6;0.7] | 2.6 | 0.7 [0.6;0.8] | 2.9 | 0.8 [0.7;1.0] | 2.3 | 0.6 [0.5;0.7] | 2.5 | 0.7 [0.5;0.8] | 2.7 | 0.7 [0.6;0.9] | |
| Non-Western | 5.8 | 1.0 | 8.7 | 1.9 [1.7;2.0] | 6.5 | 1.2 [1.2;1.3] | 13.0 | 2.4 [2.2;2.7] | 5.2 | 1.1 [0.9;1.3] | 8.7 | 2.2 [1.9;2.4] | 7.6 | 1.4 [1.2;1.5] | |
| **Marital status** | | | | | | | | | | | | | | | *** |
| Unmarried | 38.1 | 1.0 | 36.5 | 1.0 [1.0;1.0] | 41.6 | 1.2 [1.2;1.3] | 60.6 | 2.5 [2.4;2.7] | 40.3 | 1.2 [1.1;1.3] | 51.9 | 1.9 [1.8;2.0] | 37.1 | 1.0 [0.9;1.1] | |
| Married § | 61.9 | 1.0 | 63.5 | 1.0 | 58.5 | 1.0 | 39.5 | 1.0 | 59.7 | 1.0 | 48.1 | 1.0 | 62.9 | 1.0 | |

***: p<0.001

§: reference group; OR: Odds ratio compared to the reference group of being in a multimorbidity class compared to the reference class; *p*: Chi²-test for univariate association between demographic variable and classes

[1]Adjusted for age and sex

**Table 5. Demographic profile of individuals by assigned classes in the age group 16–44 years.**

| | 'No or few diseases' 90.3% § (n = 189,202) | | 'Bone-, joint diseases' 3.8% (n = 7,962) | | 'Mental illness, epilepsy' 3.5% (n = 7,378) | | 'Asthma, allergy' 1.4% (n = 2,985) | | 'Diabetes, heart diseases' 1.0% (n = 2,079) | | *p* |
|---|---|---|---|---|---|---|---|---|---|---|---|
| | n (%) | OR | n (%) | OR [95%CI][1] | n (%) | OR [95%CI][1] | n (%) | OR [95%CI][1] | n (%) | OR [95%CI][1] | |
| **Age** | | | | | | | | | | | *** |
| 16–24 § | 32.9 | 1.0 | 16.8 | 1.0 | 23.9 | 1.0 | 35.1 | 1.0 | 5.3 | 1.0 | |
| 25–34 | 34.0 | 1.0 | 27.6 | 1.6 [1.5;1.7] | 35.3 | 1.4 [1.3;1.5] | 26.3 | 0.7 [0.7;0.8] | 15.6 | 2.8 [2.3;3.5] | |
| 35–44 | 33.1 | 1.0 | 55.6 | 3.3 [3.1;3.5] | 40.8 | 1.7 [1.6;1.8] | 38.6 | 1.1 [1.0;1.2] | 79.1 | 14.7 [12.1;17.8] | |
| **Sex** | | | | | | | | | | | *** |
| Men § | 52.1 | 1.0 | 35.1 | 1.0 | 36.9 | 1.0 | 43.7 | 1.0 | 58.7 | 1.0 | |
| Women | 47.9 | 1.0 | 64.9 | 2.0 [1.9;2.1] | 63.1 | 1.9 [1.8;1.9] | 56.3 | 1.4 [1.3;1.5] | 41.3 | 0.8 [0.7;0.8] | |
| **Country of origin** | | | | | | | | | | | *** |
| Danish § | 80.5 | 1.0 | 87.3 | 1.0 | 86.1 | 1.0 | 88.1 | 1.0 | 83.8 | 1.0 | |
| Other Western | 7.4 | 1.0 | 2.8 | 0.3 [0.3;0.4] | 3.1 | 0.4 [0.3;0.4] | 8.9 | 0.4 [0.3;0.4] | 13.3 | 0.4 [0.3;0.5] | |
| Non-Western | 12.1 | 1.0 | 10.0 | 0.8 [0.7;0.8] | 10.8 | 0.8 [0.8;0.9] | 3.0 | 0.7 [0.6;0.8] | 2.8 | 1.2 [1.0;1.3] | |
| **Marital status** | | | | | | | | | | | *** |
| Unmarried | 72.0 | 1.0 | 62.2 | 1.2 [1.1;1.2] | 78.9 | 2.3 [2.1;2.4] | 69.3 | 0.9 [0.8;1.0] | 57.2 | 1.4 [1.2;1.5] | |
| Married § | 28.0 | 1.0 | 37.8 | 1.0 | 21.1 | 1.0 | 30.8 | 1.0 | 42.8 | 1.0 | |

***: p<0.001

§: reference group; OR: Odds ratio compared to the reference group of being in a multimorbidity class compared to the reference class; *p*: Chi$^2$-test for univariate association between demographic variable and classes

[1]Adjusted for age and sex.

Associations between educational attainment and multimorbidity are illustrated in Fig 2 and are also shown in S6–S8 Tables. This shows that in the age group 65+ years, shorter education was associated with higher odds of belonging to all multimorbidity classes. The trend was strongest for the class 'Many diseases' with the highest odds among those with elementary school education (OR = 2.5; 95%CI:2.2–2.7). In the age group 45–64 years, shorter education was associated with higher odds of belonging to all multimorbidity classes except 'Asthma, allergy'. The association was strongest for the class 'Many diseases' with highest odds among those with elementary school education (OR = 4.2; 95%CI:3.8;4.7). In the age group 16–44 years, shorter education was associated with higher odds of belonging to all multimorbidity classes except 'Asthma, allergy'. In Fig 3, associations between employment status and multi-morbidity are illustrated. These results are also presented in S6 and S7 Tables. In the age group 45–64 years, those who were unemployed or on early retirement had higher odds of being in any multimorbidity class compared to those who were working. Retired had higher odds of being in all multimorbidity classes except 'Asthma, allergy'. The association was especially strong for the multimorbidity classes 'Many diseases' and 'Mental illness, epilepsy'. In the age group 16–44 years, those who were unemployed or on early retirement had higher odds of being in any multimorbidity class except 'Asthma, allergy' compared to those who were working. Across age groups, those on early retirement had the highest odds of belonging to any multimorbidity class.

## Discussion

We identified seven classes among individuals in the age groups 45–64 years and 65+ years and five classes in the age group 16–44 years. Overall, the classes were similar in the three age groups, but varied in size, i.e. the class 'No or few diseases' was larger in the younger age group and the

## 65+ years

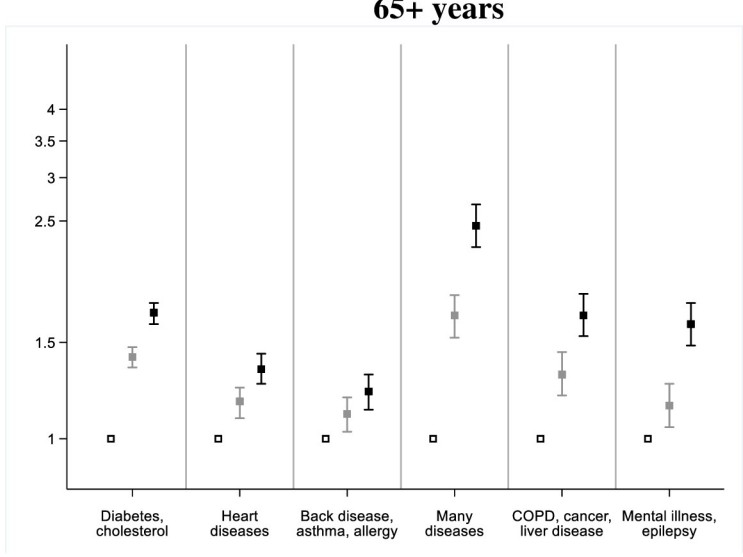

## 45-64 years

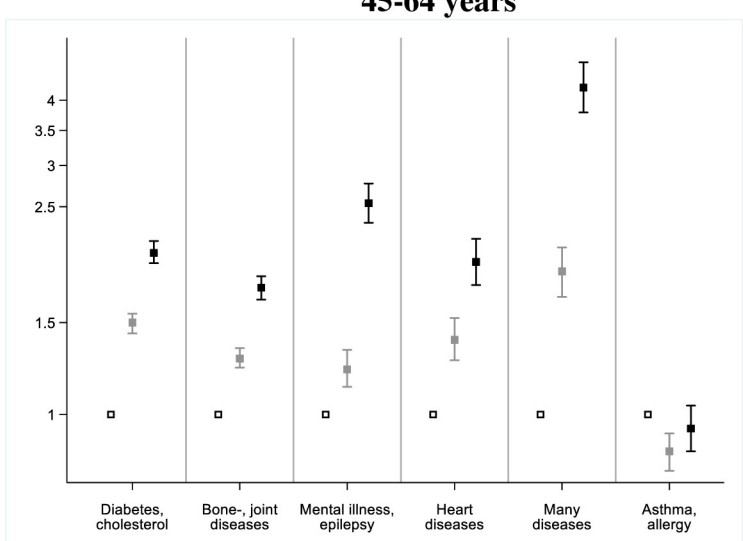

## 16-44 years

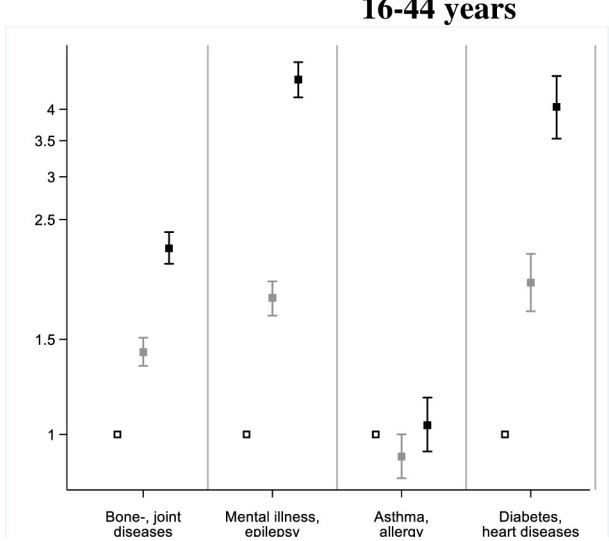

**Fig 2. Odds ratio of being in a multimorbidity class by education.** Odds ratio compared to medium/long education of being in a multimorbidity class compared to the reference class 'No or few diseases'. Odds ratios are presented on a logarithmic scale. Adjusted for age and sex. Short education: completed high school, vocational school, or short tertiary education Medium/long education: Completed medium or long tertiary education (>3 years).

majority of the multimorbidity classes were larger in the oldest age group. The class 'Many diseases' was not seen in individuals aged 16–44 years, and heart diseases and diabetes also presented in one class in this age group compared to the two different classes in the older age groups.

A previous systematic review identified three groups of patterns that were often found across studies examining patterns of multimorbidity, including one with metabolic diseases, one with mental illness, and one with musculoskeletal diseases [4]. Another systematic review identified three other groups, that partially overlapped the above, including one with cardio-metabolic disease, one with anxiety and depression, and one with pain [3]. In a third systematic review they identified three main groups characterised as one with cardiovascular and metabolic diseases, one with mental health problems, and one with allergic diseases [25]. Our results are comparable to these identified patterns of multimorbidity. However, in general it is difficult to compare our results with findings from other studies because of variations in methods, data sources and structures, populations and diseases studied. A Danish study using self-reported information on diseases also found a class with asthma and allergy as well as classes with musculoskeletal disorders, mental disorders, and cardio-metabolic disorders [6]. They however also found a class with hypertension and a smaller class with complex respiratory diseases. These differences could be influenced by differences in data sources, as some diseases may be overreported in self-reported data or underreported in register-based data. For example, allergy and many musculoskeletal diseases rarely result in hospital contacts or prescriptions that can be linked to the specific disease.

The identified patterns of multimorbidity may be consequences of underlying risk factors or of causal links between diseases that interact with each other. Unfortunately, we did not have information on possible risk factors, so it was not possible to identify if for example obesity was the underlying risk factor for being in the class 'Diabetes, cholesterol' or the class 'Diabetes, heart diseases'. Also, it is possible that the class 'COPD, cancer, liver disease' is characterised by individuals with a high prevalence of both smoking and alcohol consumption. Further studies on patterns of multimorbidity could explore possible risk factors for developing specific patterns of multimorbidity.

Our results on sociodemographic characteristics of the identified groups show that the different classes indeed differ on these characteristics. The same was previously found in a Danish study using self-reported information on diseases [6]. This highlights that it is relevant to study sociodemographic characteristics of individual patterns of multimorbidity, as differences between patterns will be overlooked when only studying characteristics of individuals with two or more diseases. The sociodemographic characteristics showed a strong association between education and employment status and odds of being in the class 'Many diseases'. The same is true for the class 'Mental illness, epilepsy'. For both classes it is possible that lower socioeconomic status is a risk factor for developing the diseases that characterise these multimorbidity classes, but it is also possible that disease development has influenced the socioeconomic status of the individual.

The results on country of origin showed that in the age group 16–44 years, individuals with non-Western origin or with non-Danish Western origin were less likely to belong to almost all the multimorbidity classes. However, in the older age groups, those with non-Western origin were more likely to belong to several multimorbidity classes compared to those with Danish origin. This could reflect a healthy migrant effect, where those who are able to migrate are the

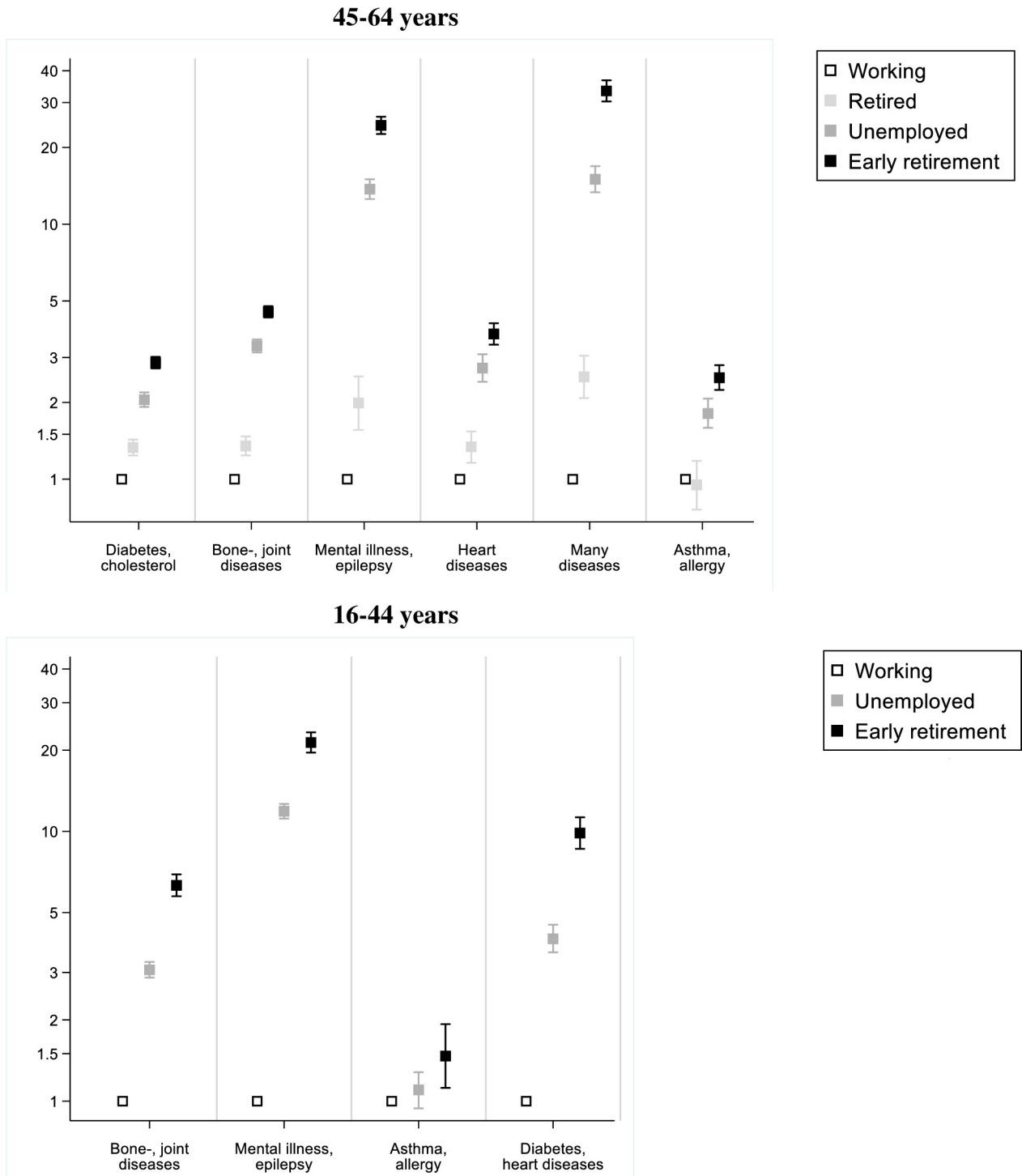

**Fig 3. Odds ratio of being in a multimorbidity class by employment status.** Odds ratio compared to medium/long education of being in a multimorbidity class compared to the reference class 'No or few diseases'. Odds ratios are presented on a logarithmic scale. Adjusted for age and sex.

healthier young people compared to the less healthy young people. However, as these healthy young immigrants grow older in Denmark, they may both adopt the lifestyles of the majority population but may also suffer from additional risk factors including psychosocial stressors

causing greater risk of developing multimorbidity. It has previously been shown that immigrants in Denmark have similar or lower risk of cardiovascular diseases, acute myocardial infarction and stroke compared to Danish-born individuals [27], but that they have higher risk of diabetes [28]. A Norwegian study has found lower rates of multimorbidity in immigrants compared to Norwegian-born individuals [29].

Our results show that there are large differences in both the sizes of the identified classes but also in the number of diseases among individuals in each class. The multimorbidity group 'Many diseases' had the largest number of diseases in individuals aged 65+ years and in individuals aged 45–64 years. This was followed by 'Mental illness, epilepsy' in the age group 65+ years and 'Heart diseases' in the age group 45–64 years. In the age group 16–44 years, the largest number of diseases was seen in the classes 'Diabetes, heart diseases' and 'Mental illness, epilepsy'. This shows that especially in the older age groups, individuals are affected by many diseases, but it also highlights that mental illness is relevant to both treatment and prevention across all age groups.

Though we did not include information on healthcare utilisation in the identified classes, the number of diseases impacts this as shown in a previous study [30]. Therefore, both the number of diseases as well as the size of the multimorbidity classes are factors that are relevant to include when making priorities on prevention and treatment. For example, in the age group 65+ years, the class 'Diabetes, cholesterol' is quite large, while the class 'Many diseases' is smaller but is characterised by individuals with a high number of diseases.

The study design and data sources chosen for this study entailed several strengths and limitations. The strengths include a large sample size and a register-based study population, which implies that selection into the study population is avoided. Also, as the diseases are based on register-based information from nationwide registers, information bias is low and the validity is high when the disease causes use of hospital use or prescription drugs [31]. Finally, due to the large sample size it was possible to analyse on three separate age groups enabling identification of age specific patterns of multimorbidity. The limitations of the study include that diseases are not included if they have only been treated in primary care and no disease specific treatment drugs have been prescribed. Therefore, diseases such as many mental illnesses, allergy, and musculoskeletal diseases may be under reported. However, we tried to take this into account by including information on prescriptions which should allow for at least partly detection of some of these diseases, but the validity of this information may be lower than that of ICD-10 codes. Another limitation is in the lack of clear temporality between socioeconomic factors and disease development, which does not allow for inferences on the direction of identified associations. A third limitation is that the method of assigning individuals to the class with the highest probability could result in biased estimates in the multinomial logistic regression analyses. The sensitivity analysis applying weights for the probability of belonging to the assigned class showed only small differences in results between the two groups, but it is possible that this method does not adequately address bias introduced in the analysis. Finally, this study does not include information on associations between multimorbidity and possible consequences such as hospital admissions, employment, and mortality, which could help inform intervention strategies.

In conclusion, we identified multimorbidity classes for three age groups in Denmark and found that there were social inequalities in odds of belonging to a multimorbidity class. These social inequalities varied but were especially strong in the classes named 'Many diseases' and 'Mental illness, epilepsy'. Future studies could examine lifestyle related risk factors for developing the specific multimorbidity classes, as well as the consequences such as hospital admissions, employment, and mortality, specific to each multimorbidity class. The findings also highlight that healthcare systems need to be able to accommodate complex disease pictures among patients and that these may be associated with social circumstances which should also be taken into account in the treatment of these patients.

## Supporting information

**S1 Table. Included diseases in latent class analyses.**
(DOCX)

**S2 Table. Disease prevalence in classes in the age group 65+ years.**
(DOCX)

**S3 Table. Disease prevalence in classes in the age group 45–64 years.**
(DOCX)

**S4 Table. Disease prevalence in classes in the age group 16–44 years.**
(DOCX)

**S5 Table. Fit statistics for analyses in the three age groups.**
(DOCX)

**S6 Table. Educational level of individuals by assigned classes in the age group 65+ years.**
(DOCX)

**S7 Table. Educational level and employment status of individuals by assigned classes in the age group 45–64 years.**
(DOCX)

**S8 Table. Educational level and employment status of individuals by assigned classes in the age group 16–44 years.**
(DOCX)

**S9 Table. Demographic profile of individuals by assigned classes in the age group 45–64 years.** Results weighted for probability of class membership.
(DOCX)

## Author Contributions

**Conceptualization:** Sanne Pagh Møller, Bjarne Laursen, Caroline Klint Johannesen, Janne S. Tolstrup, Stine Schramm.

**Data curation:** Bjarne Laursen.

**Formal analysis:** Sanne Pagh Møller, Bjarne Laursen.

**Methodology:** Sanne Pagh Møller, Bjarne Laursen, Caroline Klint Johannesen, Janne S. Tolstrup, Stine Schramm.

**Supervision:** Stine Schramm.

**Writing – original draft:** Sanne Pagh Møller.

**Writing – review & editing:** Sanne Pagh Møller, Bjarne Laursen, Caroline Klint Johannesen, Janne S. Tolstrup, Stine Schramm.

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
