## [Decision Letter · Decision Letter 0]

2 Mar 2020

PONE-D-20-01107

Pattern of multimorbidity and demographic profile of latent classes in a Danish population – a register-based study

PLOS ONE

Dear Ms Møller,

Thank you for submitting your manuscript to PLOS ONE. After careful consideration, we feel that it has merit but does not fully meet PLOS ONE’s publication criteria as it currently stands. Therefore, we invite you to submit a revised version of the manuscript that addresses the points raised during the review process.

We would appreciate receiving your revised manuscript by Apr 16 2020 11:59PM. To enhance the reproducibility of your results, we recommend that if applicable you deposit your laboratory protocols in protocols.io, where a protocol can be assigned its own identifier (DOI) such that it can be cited independently in the future. For instructions see: http://journals.plos.org/plosone/s/submission-guidelines#loc-laboratory-protocols

A **rebuttal letter** that responds to **EACH** point raised by the academic editor and reviewer(s). This letter should be uploaded as separate file and labeled 'Response to Reviewers'.A **marked-up copy** of your manuscript that highlights changes made to the original version. This file should be uploaded as separate file and labeled 'Revised Manuscript with Track Changes'.An **unmarked version** of your revised paper without tracked changes. This file should be uploaded as separate file and labeled 'Manuscript'.

We look forward to receiving your revised manuscript.

Kind regards,

Brecht Devleesschauwer

Academic Editor

PLOS ONE

Additional Editor Comments (if provided):

In your revision note, please include EACH comment of the reviewers, provide your reply, and when relevant, include the modified/new text (or motivate why you decided not to modify the text). Note that failure to do so may result in a rejection of the manuscript.

Journal Requirements:

2. In ethics statement in the manuscript and in the online submission form, please provide additional information about the patient records used in your retrospective study. Specifically, please ensure that you have discussed whether all data were fully anonymized before you accessed them and/or whether the IRB or ethics committee waived the requirement for informed consent. If patients provided informed written consent to have data from their medical records used in research, please include this information.

3. According to our policies, if materials, methods, and protocols are well established, authors may cite articles where those protocols are described in detail, but the submission should include sufficient information to be understood independent of these references (https://journals.plos.org/plosone/s/submission-guidelines#loc-materials-and-methods). Please ensire that you provide more information on the methodology used, and that the algorithms applied are reported.

Reviewers' comments:

Reviewer's Responses to Questions

**Comments to the Author**

1. Is the manuscript technically sound, and do the data support the conclusions?

Reviewer #1: Yes

Reviewer #2: Yes

Reviewer #3: Yes

2. Has the statistical analysis been performed appropriately and rigorously? 

Reviewer #1: No

Reviewer #2: Yes

Reviewer #3: Yes

3. Have the authors made all data underlying the findings in their manuscript fully available?

Reviewer #1: No

Reviewer #2: Yes

Reviewer #3: Yes

4. Is the manuscript presented in an intelligible fashion and written in standard English?

Reviewer #1: Yes

Reviewer #2: Yes

Reviewer #3: Yes

5. Review Comments to the Author

Reviewer #1: General remark: The aim of this study is to identify and describe demographic characteristics of multimorbidity classes in three age groups by applying latent class analysis on registry data from a random sample of the Danish population. I find the study interesting and relevant, but there are a number of methodological weaknesses in the study that I think should be addressed.

Page 1, line 1

I suggest that the authors consider replacing 'pattern' with 'patterns'.

Page 3, line 50-53

This section is difficult to understand. While knowledge about LCA is still limited among many epidemiologists and other health researcher, I suggest that the authors use a more straightforward and understandable explanation of LCA. See, e.g. Mariano Porcu, Francesca Giambona (2017). Introduction to Latent Class Analysis With Applications. The Journal of Early Adolescence, 37, 1: 129-158, or some of the standard literature on LCA (e.g. Bartholomew, Knott, & Moustaki (2011) or McCutcheon (1987)).

Page 3, line 54-55

'This technique has been used in previous studies identifying patterns of multimorbidity in different populations (6-8).' The authors should add this study to the reference list:

Park B, Lee HA, Park H. Use of latent class analysis to identify multimorbidity patterns and associated factors in Korean adults aged 50 years and older. PLoS One. 019;14(11):e0216259. Published 2019 Nov 13. doi:10.1371/journal.pone.0216259

Page 4, line 75

Which register was the sample extracted from?

Page 4, line 76

How is 'permanent residence in Denmark' defined?

Page 5, line 107

'LCA was carried out separately for three age groups…' A major disadvantage of making three instead of one latent class model is that one cannot compare the prevalence of latent classes across age because the classes are specific to the age groups.

In my opinion, it is a better analytical strategy to estimate a latent class model for the entire sample and either include age in the model at par with the other indicators or, even better, include age as a covariate (cf. Dayton, C. M., & Macready, G. B. (1988). Concomitant variable latent class models. Journal of the American Statistical Association, 83, 173-178). The latter type of model allows you to answer the question: Is age predictive of class membership? A third way to proceed with this is to use a so-called three-step procedure, cf. below.

Page 6, line 115-117

'To asses associations between the demographic variables and the different classes, we applied a multinomial logistic regression model, which was adjusted for age and sex.' Here a three-step procedure is utilized within each of the three LCA models.

This involves (1) estimating a LCA model, using only data on the indicators of the latent classes, (2) assigning a latent class to each individual based on the model from step 1, and (3) estimating a logistic regression model with the assigned values from step 2 as dependent variable.

The three-step method has the flaw that the values assigned in its second step are not equal to the true values of the latent classes as defined by the first step (the assignment to latent classes is probabilistic). This creates a measurement error (misclassification) problem which means that the third step will yield biased estimates of the regression model.

Several methods have been developed to correct for this bias (cf. Bakk, Zs., Tekle, F.B., and Vermunt, J.K. (2013). Estimating the association between latent class membership and external variables using bias adjusted three-step approaches. Sociological Methodology, 43, 272-311). My suggestion to the authors is that they apply one of these bias-adjusted methods.

Page 7, table 1

How is 'Short education' and 'Medium/long education' defined?

Page 8, line 133-143

When conducting an LCA, a major decision in the process of model specification is choosing the number of latent classes to retain. On page 5, lines 103-106, the six criteria that were used to select the number of latent classes are mentioned. In order to allow the reader to assess the quality of the model chosen, I suggest that the authors in this section briefly describe how the six criteria have been applied and weighed against each other. How well does the selected model fit the data according to the criteria? How good is the latent class separation? Etc. The authors might consider adding a supplementary table with more detailed information.

Page 14, line 22

'Overall, the classes were similar in the three age groups, but varied in size…' I agree with this finding, which shows that nothing is really gained by using three models rather than one.

Reviewer #2: This paper presents the pattern of multimorbidity and the associated demographic profile among a representative sample of Danish population using a register-based data.

With the ageing of the population, multimorbidity is an important public health problem and epidemiological data on this topic, is therefore highly relevant. Even though other studies have investigated the pattern of multimorbidity, this paper is interesting and adds useful information to the existing literature. The study is based on a large random sample of Danish population. The use of register-based information and separate age-groups analyses are other strengths of the study. The authors used appropriate methods to explore the research questions. I only have a few minor remarks.

Introduction

Line 43-44 : The authors state that multimorbidity is a public health burden today and is estimated to increase in the future. There is lack of data to support this statement. It will be useful to provide some information, e.g., prevalence of multimorbidity worldwide and/or in Denmark.

The sentence “Apply methods to assess multimorbidity include…..” should be moved in the method section. Furthermore, authors should justify why LCA has been chosen instead of other methods.

Methods

Please clarify how the 47 chronic diseases were clinically validated. Those the diseases selected based on their ICD-10 codes or others?

Line 84 - 86, it is mentioned that “Identification of diseases was made by applying diagnoses and purchase of prescribed medications primarily based on algorithms applied in previous studies…”. In my opinion, it is important to provide more detailed on those algorithms in order to facilitate readers comprehension. An overview of chronic diseases is presented in additional table S1 and it is mentioned that the algorithms will also find in the same table. However, I miss the later in table S1, please address this.

Referring to my previous remark, information on the methods used to identify the pattern of multimorbidity should be address in this section. Furthermore, please motivate the use of LCA compared to other methods.

A large number of sociodemographic covariates are included in the study. However, I find it weird that a univariate logistic regression model is used to asses associations between the demographic variables and the different classes instead of multivariate one. Why? This may affect interpretation of the results.

Results

As one of the strength of the present study in contrast to previous studies, is the identification of age specific patterns of multimorbidity, it would be useful to provide in the table 2 the mean number of diseases by age-group, not only by classes of diseases. Such information could also point out in what extend multimorbidity can differ across age-groups and why age-specific analyses are useful.

Discussion:

Compared to other studies, authors found a large range of classes. How they can justify this finding?

How those the authors interpret the fact that they found similar classes across age-group?

Do this meant youngest and older people suffer from the same chronic diseases? problem of misclassification?

Conclusions

In the conclusion, a summary of the results is presented and ideas for future researches. I miss implications of the study for public health and health policymakers.

References

An important study has been carried out in Denmark in the same area which is lacking in this article. Authors should consider adding this reference at least in the introduction and or discussion:

Breinholt Larsen F, Hauge Pedersen M, Friis K, Glümer C, Lasgaard M. Patterns of Multimorbidity in the General Danish Population. A Latent Class Analysis. European Journal of Public Health. 2018 Nov 1;28(suppl_4):cky214-062.

Reviewer #3: The authors have conducted a sound analyses of a large sample of high quality Danish data (~470,000 individuals) in order to characterize detailed multimorbidity profiles across a sample of the population. Determining multi morbidity disease profiles and the underlying demographics which may be associated with those profiles is of interest to the medical community and is very topical. The link with educational data is strong and unique - as not many other countries are able to link such data to healthcare information at this sacale. Whilst the current paper uses appropriate and advanced latent class analyses techniques to define multi morbidity clusters from 47 individual diseases - the main contribution of the study is largely descriptive as there are no analyses of the outcomes associated with different MM clusters. Without information on the impact of these newly define MM clusters it is not clear what the clinical impact or recommendation for changes to healthcare is on the basis of this research. Whilst it is interesting to see that females are more at risk of certain MM presentations compared to males for example - the potential for targetted intervention for prevention of these MM presentations in females will only be made if we know whether these MM clusters are particularly important in terms of quality of life or mortality burden for example. The Academy of Medical Sciences report on multimorbidity clearly highlights the lack of studies looking at MM and association with outcomes as a key problem and knowledge gap. This limitation should be made clear - if not able to be addressed with the current data.

There are a number of additional aspects which the authors need to consider should the paper be accepted for publication as outlined below:

1. Missing key references in the introduction for work which has used LCA to assess MM clusters in detail - including a number of references found in the AMS report, particularly the work by Hall et al 2018 in PloS Medicine using LCA to look at clusters of multimorbidity and long term outcomes.

2. The authors claim the risk of selection bias in the study is limited because this is a registry based analyses. However - the data included represents a 10% sample of the population - without clear and detailed inforamtion alongside table 1 of the characteristics of the full population it is impossible to say whether or not these 10% are indeed representative. More information on this in needed to support the claim.

3. What is the justification / reasoning for the three specific age groups chosen? Are these based on particular clinically relevant age cut offs/made around healthcare service provision? please make clear throughout.

4. I could not find any information about the choices for the specific clusters - the authors state in the methods this was made based on model fit statistics as well as how meaningful the emerging classes were - however provide no tables of model fit statistics and discussion of class cut offs. This is important information to include - to ensure scientific rigour and transparency in the decision making process. Please also clarify the maximum number of clusters that were checked.

5. Disease definitions - more detailed is required on which conditions, or how many individuals, were identified as having a particular disease through the prescription of medications only. Was this done only for certain prescriptions where the medication would only ever be prescribed for that particular disease ? Many drugs will be used to treat a number of conditions, therefore I question the accuracy of a 'diagnoses' being inferred based on prescription of one medication twice. A sensitivity analyses of classes and disease profiles which excludes the prescription only 'diagnoses' is strongly advised so that the impact of this decision on the final outcomes can be determined.

6. The paper needs more detailed information with regards to the make up of the clusters - this is provided in tables in the supplement, but needs to be presented up front in the main paper as without it the over arching cluster descriptions can't be easily interpreted. Ideally - the authors should try to determine the most accessible way to present this high level of detailed information visually or graphically rather than in tabular format.

7. Please ensure all tables are fully annotated with additional inforamtion in footnotes. In table 1 - please clarify the full definition of "sick leave etc.". Also in Table 1 please add a footnote to say what the '-' reefers to - i.e. is this missing data or not applicable (if the latter - why is it not applicable for a over 65 year old to be on sick leave?). For marital status - please be more specific. Current/ever/at baseline?).

6. PLOS authors have the option to publish the peer review history of their article (what does this mean?). If published, this will include your full peer review and any attached files.

Reviewer #1: No

Reviewer #2: No

Reviewer #3: No

---

## [Author Response · Author response to Decision Letter 0]

29 Apr 2020

We have addressed the reviewer comments in the attached response letter.

---

## [Decision Letter · Decision Letter 1]

10 Jun 2020

PONE-D-20-01107R1

Patterns of multimorbidity and demographic profile of latent classes in a Danish population – a register-based study

PLOS ONE

Dear Dr. Møller,

Thank you for submitting your manuscript to PLOS ONE. After careful consideration, we feel that it has merit but does not fully meet PLOS ONE’s publication criteria as it currently stands. Therefore, we invite you to submit a revised version of the manuscript that addresses the points raised during the review process.

A **rebuttal letter** that responds to each point raised by the academic editor and reviewer(s). You should upload this letter as a separate file labeled 'Response to Reviewers'.A **marked-up copy** of your manuscript that highlights changes made to the original version. You should upload this as a separate file labeled 'Revised Manuscript with Track Changes'.An **unmarked version** of your revised paper without tracked changes. You should upload this as a separate file labeled 'Manuscript'.

We look forward to receiving your revised manuscript.

Kind regards,

Brecht Devleesschauwer

Academic Editor

PLOS ONE

Additional Editor Comments (if provided):

Thank you for addressing the reviewer comments. Reviewer #1 raised a final concern about the overall clarity of the manuscript, which could be addressed in a final, minor revision round.

Reviewers' comments:

Reviewer's Responses to Questions

**Comments to the Author**

1. If the authors have adequately addressed your comments raised in a previous round of review and you feel that this manuscript is now acceptable for publication, you may indicate that here to bypass the “Comments to the Author” section, enter your conflict of interest statement in the “Confidential to Editor” section, and submit your "Accept" recommendation.

Reviewer #1: (No Response)

Reviewer #2: All comments have been addressed

Reviewer #3: All comments have been addressed

2. Is the manuscript technically sound, and do the data support the conclusions?

Reviewer #1: Yes

Reviewer #2: Yes

Reviewer #3: Yes

3. Has the statistical analysis been performed appropriately and rigorously? 

Reviewer #1: No

Reviewer #2: Yes

Reviewer #3: Yes

4. Have the authors made all data underlying the findings in their manuscript fully available?

Reviewer #1: No

Reviewer #2: (No Response)

Reviewer #3: Yes

5. Is the manuscript presented in an intelligible fashion and written in standard English?

Reviewer #1: Yes

Reviewer #2: Yes

Reviewer #3: Yes

6. Review Comments to the Author

Reviewer #1: Line 96-99: "A third limitation is that the method of assigning individuals to the class with the highest probability could result in biased estimates in the multinomial logistic regression analyses. This bias seemed to influence the estimates for class membership in the smaller classes, where the presented results would be conservative estimates of associations."

It is not possible for the reader to assess how the authors reached this conclusion. Have they done further analysis with another method? If so, which one? And if so, why didn't they report these results instead? I reiterate my recommendation from the first review to use a bias-adjusted method to investigate the association between multimorbidity classes and sociodemographic characteristics.

General comment: I still find this to be a worthwhile study. But the information it provides is very complex. No less than 19 latent patterns of disease have been identified within three age groups. The authors have gone a long way in communicating the results in graphs and tables, but most readers will have difficulty grasping the significance of the individual disease classes. Consider if this could be made clearer.

Reviewer #2: Thank you for this revised version of the manuscript and for addressing my concerns. I recommend the manuscript for publication.

Reviewer #3: (No Response)

7. PLOS authors have the option to publish the peer review history of their article (what does this mean?). If published, this will include your full peer review and any attached files.

Reviewer #1: Yes: Finn Breinholt Larsen

Reviewer #2: No

Reviewer #3: No

---

## [Author Response · Author response to Decision Letter 1]

24 Jul 2020

Reviewer comments have been adressed in the Rebuttal letter.

---

## [Editor Report · Decision Letter 2]

27 Jul 2020

Patterns of multimorbidity and demographic profile of latent classes in a Danish population – a register-based study

PONE-D-20-01107R2

Dear Dr. Møller,

We’re pleased to inform you that your manuscript has been judged scientifically suitable for publication and will be formally accepted for publication once it meets all outstanding technical requirements.

Kind regards,

Brecht Devleesschauwer

Academic Editor

PLOS ONE
---

## [Editor Report · Acceptance letter]

30 Jul 2020

PONE-D-20-01107R2 

Patterns of multimorbidity and demographic profile of latent classes in a Danish population – a register-based study 

Dear Dr. Møller:

I'm pleased to inform you that your manuscript has been deemed suitable for publication in PLOS ONE. Congratulations! Your manuscript is now with our production department. 

Kind regards, 

on behalf of

Prof. Dr. Brecht Devleesschauwer 

Academic Editor

PLOS ONE